# Staging Fibrosis in Chronic Viral Hepatitis

**DOI:** 10.3390/v14040660

**Published:** 2022-03-23

**Authors:** Ana Carolina Cardoso, Claudio Figueiredo-Mendes, Cristiane A. Villela-Nogueira, Patrick Marcellin

**Affiliations:** 1Postgraduate Internal Medicine Program, Hepatology Division, Clementino Fraga Filho University Hospital, School of Medicine, Federal University of Rio de Janeiro, Rio de Janeiro 21941-617, Brazil; 2Hepatology Division, General Hospital, Santa Casa da Misericórdia do Rio de Janeiro, Rio de Janeiro 20020-022, Brazil; claudiofigueiredomendes@gmail.com; 3Internal Medicine Department, Hepatology Division, School of Medicine, Federal University of Rio de Janeiro, Rio de Janeiro 21941-617, Brazil; crisvillelanog@gmail.com; 4Hepatology Department, Hôpital Beaujon, APHP, INSERM, University of Paris, 92110 Clichy, France; patrick.marcellin@aphc.org

**Keywords:** fibrosis, chronic hepatitis, hepatitis B, hepatitis C, non-invasive

## Abstract

Staging fibrosis accurately has always been a challenge in viral hepatitis and other liver diseases. Liver biopsy is an imperfect gold standard due to its intra and interobserver agreement limitations and additional characteristics such as its safety and cost. Hence, non-invasive tests have been developed to stage liver fibrosis. In addition to serological biomarkers, physical tests with reasonable accuracy are available and adopted in the daily clinic regarding viral hepatitis fibrosis staging. In this review, we discuss the published data regarding the staging of liver fibrosis in chronic hepatitis B and C, emphasizing non-invasive markers of fibrosis, both serological and physical. Moreover, we also discuss a persistent central gap, the evaluation of liver fibrosis after HCV cure.

## 1. Introduction

Chronic viral hepatitis B and C, despite the therapeutic advances seen in recent years, continue to be an important public health problem worldwide. In this scenario, the characterization of inflammation and fibrosis observed in chronic viral infections is mandatory. Among the histological modifications, the definition of the degree of fibrosis is of great prognostic importance. Almost exclusively, liver biopsy was used to evaluate histological changes in chronic viral hepatitis for a long time. However, the risks and costs of an invasive procedure such as liver biopsy led to the search for alternatives to assess hepatic fibrosis. Identifying several direct and indirect markers (biological or physical) with adequate accuracy has modified the clinical practice.

The biological non-invasive methods include direct and indirect serum biomarkers such as scoring systems that involve routine laboratory and patient parameters. The non-invasive physical methods are image-related for diagnosing fibrosis, such as transient and ultrasound elastography and magnetic resonance. Both biological, direct or indirect, and physical markers show good accuracy but should be used with caution in certain situations, such as in the presence of inflammatory processes or cholestasis. This review aims to discuss the data regarding the staging of liver fibrosis, emphasizing non-invasive markers in patients with chronic hepatitis C (CHC) and B (CHB).

## 2. Chronic Hepatitis C

Hepatitis C virus (HCV) infection directly modulates signaling and metabolic pathways by viral proteins. Additionally, it indirectly induces host antiviral immune responses leading to chronic inflammation and liver fibrogenesis [1].

As in other liver diseases, fibrosis is a precursor of cirrhosis and the main predictor of liver-related morbidity and mortality regarding portal hypertension and hepatocellular carcinoma.

In CHC, liver biopsy is no longer used as a tool to stage fibrosis due to the safety issues and high cost, among others. Overall, non-invasive tests (NITs) are cost-effective tests that help estimate fibrosis with high accuracy [2,3], mainly in pre-treatment patients. The gold standard for the validation of NITs has been liver biopsy. The histological analysis is performed according to several scores. The most used are Ishak [4] and, currently, METAVIR score [5]. According to the METAVIR scoring system [5], septal fibrosis is defined as significant (F2), and bridging fibrosis or cirrhosis as advanced (equal or greater than F3); F4 equals to cirrhosis.

NITs are classified into two major groups, biological and physical/radiological. Biological NITs include indirect and direct biomarkers, and the physical ones comprise elastography methods [2]. Physical tests are new techniques such as transient hepatic elastography. The first one was available by Fibroscan equipment. Point-shear wave elastography or 2D-shear wave elastography are additional ultrasound types of equipment with specific elastography software. In addition, there is also magnetic Resonance elastography that, although very accurate, is still costly and demands a more complex structure to be used in daily routine. Both serological and physical NITs may be used alone or in combination to better stage fibrosis replacing liver biopsy, an imperfect gold standard. The perfect NIT does not exist so far, but the characteristics that fulfill the main requirements for the best NIT are displayed in Table 1. Of note, in many patients with clinical cirrhosis, an ultrasound is adequate to establish the diagnosis of cirrhosis. Additional tests are usually unnecessary, except for the stratification of portal hypertension, which we will discuss further.

Strikingly, after treatment with a direct-acting antivirals (DAAs) and HCV virological cure, there are still some unanswered questions regarding fibrosis staging and the value of NITs as prognostic tools in CHC. This will also be discussed later.

The World Health Organization aims to eliminate HCV by 2030, but this is still a challenge even in developed countries. It is essential to diagnose and treat at least 80% of HCV-infected patients to reach this goal. Currently, the primary strategy regarding fibrosis staging is to identify patients with advanced fibrosis/cirrhosis. Generally, the diagnostic performance of serum markers is better for cirrhosis than significant fibrosis in CHC [6]. These data are very relevant in the era of DAA treatment. Accurate fibrosis staging before treatment is less clinically crucial than identifying advanced fibrosis or cirrhosis, which requires continued post-treatment surveillance. These patients still need follow-up after SVR regarding hepatocellular carcinoma (HCC), portal hypertension, and cirrhosis complications screening [7]. Moreover, defining the diagnosis of cirrhosis before treatment is of utmost importance. Furthermore, it is also necessary to stratify those with decompensating cirrhosis since this group will need additional stratification to start treatment before or after liver transplant [7].

### 2.1. Biological Non-Invasive Tests

#### 2.1.1. Indirect Tests

Indirect biomarkers are cost-effective and straightforward scores frequently used in daily clinics to stratify liver fibrosis in CHC. The most used are the AST to platelet ratio index [6] ([APRI = (AST/upper limit of normal) × 100/platelet count]) and Fibrosis Index-Based in Four Factors ([FIB4 = (age × AST)/(Platelets × √ALT) [8] scores. Their main limitation is a gray zone that cannot accurately define if the patient has significant fibrosis. One may need a complementary test to stratify the patient better in this situation. However, as of today, the most important issue is the identification of patients with advanced fibrosis or cirrhosis, and these tests are suitable to use in clinical practice. The pooled area under the receiving operating characteristic (AUROC) for cirrhosis for an APRI > 2.0 was 0.84 with a specificity of 94% [6]. In the original study that included patients with HIV-HCV coinfection, FIB-4 identified advanced fibrosis (F ≥ 3) with an AUROC curve of 0.76 and a specificity of 97% for a cutoff of 3.25 [9]. However, FIB-4 was further validated in 592 patients with CHC [8]. In this study, a value > 3.25 had an AUROC of 0.91 for cirrhosis. A value < 1.45 had a sensitivity of 74% for excluding advanced fibrosis and a negative predictive value for advanced fibrosis of 94.7% [8]. These results have been corroborated in additional studies, showing that FIB-4 with a cutoff > 3.25 is a useful NIT for diagnosing advanced fibrosis [10,11].

#### 2.1.2. Direct Tests

The direct biomarkers include products that participate in different pathways of the hepatic fibrogenesis process. They include hyaluronic acid (HA), a glycosaminoglycan [12,13], glycoproteins such as laminin [14], and YKL-40 [15]; products from the collagens family, such as procollagen III N-peptides (PIIINP) [12] and type IV collagen [16]; and collagenases and their inhibitors, such as matrix metalloproteases (MMP) [17] and tissue inhibitory metalloprotease-1 (TIMP-1) [18]. Most of these direct markers have been developed and studied in the era of interferon-based treatment for CHC. At that time, due to the several side effects of interferon-based therapy, it was essential to define the fibrosis stage of the patients accurately. Hence, patients with mild or no fibrosis could wait for better treatments in the future. Currently, some of the direct tests were grouped as panel scores, such as the patented enhanced liver fibrosis score (ELF) that comprises levels of HA, TIMP1, and PIIINP [19]. The ELF score has been validated in CHC, with an AUROC of 0.85, for advanced fibrosis (F3) [20].

Other patented panels have also been validated for scoring fibrosis in CHC that mix direct with indirect tests. The most validated are Fibrotest [21], Fibrometer [22], and Hepascore [23]. Fibrotest showed an AUROC of 0.87 for significant fibrosis with a 0.48 cutoff. For cirrhosis, the AUROC was 0.82 for a cutoff of 0.74.

A prospective comparison of six non-invasive scores (MP3, Fibrometer^®^, Fibrotest^®^, Hepascore^®^, Forns, and APRI) for the diagnosis of liver fibrosis in CHC demonstrated an AUROC of 0.91 for Fibrometer to 0.76 for FORNS for discriminating advanced fibrosis [24].

### 2.2. Physical Tests

Although indirect serum biomarkers have been available and used for decades, during the last twenty years, ultrasound- and resonance-based techniques have become known as a cornerstone for stratifying fibrosis with good accuracy due to elastography techniques [25]. These methods have been extensively validated in patients with CHC.

The first ultrasound-based method was vibration-controlled transient elastography (Fibroscan^®^, Echosens, Paris, France) (VCTE). It is currently considered the point-of-care tool for diagnosing fibrosis in HCV-infected patients and other etiologies of liver diseases. VCTE is easy, safe, reproducible, and accurate for diagnosing liver fibrosis in CHC.

However, it has some drawbacks. If the elevation in ALT is higher than five times the upper limit of normality, significant necro-inflammatory activity, alcohol excess, amyloidosis, or other comorbidities that lead to hepatic congestion such as right heart failure or cholestasis will significantly elevate liver stiffness measure (LSM) and impact the reliability of the correct LSM. Hence, in the face of these situations and the presence of ascites, VCTE should not be performed. Additionally, if the patient LSM cutoffs for significant fibrosis and cirrhosis have been widely validated for CHC: ≥7.1 kPa for F2; ≥9.5 kPa for F3 and ≥12.5 kPa for F4. So far, when the XL probe is used in CHC, the same thresholds are adopted. Notably, fibrosis stages may overlap with liver fibrosis’s intermediate stages in CHC. This overlapping must be considered when analyzing the results of VCTE and other US-based elastography methods.

Point-shear wave (PSW-E) and two-dimensional shear-wave (2D-SWE) elastography are additional ultrasound-based techniques available, although less validated when compared to VCTE for the staging of liver fibrosis in CHC. They have the advantage of making possible the visualization of the liver tissue and vessels since the software that performs the elastography is incorporated into ultrasound equipment. On the other hand, these types of equipment have other limitations that may impact its reliability even in CHC. Various kinds of equipment perform PSW-E and also 2D-SWE. Furthermore, there is a lack of uniformity in commercial system design, variability in shear wave frequency, sampling rates, and other technical parameters that limit the comparison of LSM across manufacturer systems. It also requires a more extended training period and a trained radiologist to perform the exam.

Although it has good accuracy for diagnosing cirrhosis, magnetic resonance is limited by its complexity and cost. It is also not widely available for fibrosis staging in daily clinics. Its primary applicability in CHC is the diagnosis of HCC when a suspect nodule is visible by ultrasound in patients with HCV-related cirrhosis.

### 2.3. Evaluation of Fibrosis in Post-SVR Patients

There is a high frequency of CHC patients who have been cured after treatment with DAAs. Although NITs are readily available tools, they are still not approved for use either during treatment or after SVR. Several studies have shown a decrease in elastography results in patients with SVR. This decrease might be the consequence of improving inflammatory activity after cure. In the interferon era, a reduction in liver stiffness correlated with treatment response in longitudinal studies [26,27]. Even in liver cirrhosis, the cutoff of 12 kPa was too low to diagnose cirrhosis after SVR in a study in cirrhotic with paired liver biopsies before and after SVR [28].

Moreover, additional comorbidities such as type 2 Diabetes Mellitus may influence liver stiffness after treatment, even in patients with SVR [29]. This way, NITs (serum biomarkers, VCE, p-SWE, 2D-SWE, and MRE) are currently not recommended to detect fibrosis regression after SVR in CHC. Of note, there are no reliable cutoffs approved for staging fibrosis after SVR, and the values adopted before SVR should not be used. After SVR, there is a significant improvement in inflammation, and the results obtained from the NITs could reflect only this improvement and not of fibrosis. Some studies suggest that a cutoff of 12 kPa would reflect advanced liver disease after SVR. However, new longitudinal studies with a longer follow-up and more included patients are still necessary to validate accurate cutoffs for staging fibrosis in post-SVR patients.

Of note, NITs may be helpful in post-SVR to evaluate prognosis regarding liver-related outcomes in CHC. A VCTE threshold of 21 kPa for clinically significant portal hypertension after SVR has been suggested for these patients.

Regarding clinical significant portal hypertension (CSPH), the BAVENO consensus indicated elastography would be a valuable tool to discard esophageal varices that needed intervention if patients have platelets < 150.000/mm^3^) and liver stiffness by VCTE under 20 kPa [33].

In the same way, VCTE values above 30 kPa or a FIB-4 above 9.0 in the baseline before treatment suggests a higher risk of HCC after SVR [30]. Additionally, a LSM decrease of less than 30% by VCTE values before and after the end of treatment would also be an independent risk for HCC in the future [31]. Although further studies are needed to understand better the prognostic value of NIT tests after SVR, it might be reasonable to perform NIT yearly in SVR patients [32].

In conclusion, fibrosis staging in CHC patients is currently performed with NITs that are widely available. Most NITs are easily achieved and easy to interpret, and their choice depends on the availability of each center. Liver biopsy in HCV is seldom used except regarding additional diagnoses such as autoimmune diseases of the liver or a necessity of inclusion in clinical studies for new drugs such as in HCV cured patients with NASH. The cutoffs after SVR are still unknown, and NITs are not validated for staging fibrosis in these specific groups.

## 3. Chronic Hepatitis B

Hepatitis B infection (HBV) is a worldwide endemic health problem with many affected individuals. The World Health Organization estimates 296 million people were living with chronic hepatitis B in 2019, with 1.5 million new infections each year [34]. Following the acute infection, patients may progress to chronic disease and cirrhosis with liver-related complications such as hepatocellular carcinoma [35,36]. Therefore, assessing the severity of the liver disease is a mandatory approach for these patients to identify those with advanced liver disease for surveillance and treatment indication. Complimentary evaluations such as abdominal ultrasound and laboratory tests should also be performed. In this scenario, the staging of liver fibrosis can be assessed by liver biopsy or non-invasive markers [37,38]. Non-invasive markers in chronic hepatitis B individuals are now routinely used and are addressed in numerous guidelines [25,32,33,39,40] and showed in Table 2.

### 3.1. Biological Non-Invasive Tests

Serum biomarkers of fibrosis are well-validated in patients with chronic viral hepatitis (with more evidence for CHC than for chronic hepatitis B and HIV-HCV coinfection) [6,7,8,9]. Well known for its use in HCV, aspartate aminotransferase to platelet ratio index [10] was evaluated in a meta-analysis including nine studies and 1798 chronic HBV patients (CHB). The AUROC of APRI for significant fibrosis and cirrhosis were 0.79 and 0.75, respectively, which suggests that APRI shows limited value in identifying hepatitis B-related significant fibrosis and cirrhosis [41]. In a study of Lin et al. [42], 631 patients with chronic HBV, APRI and fibrosis 4 score-index (FIB-4 = age (year) × AST [U/L]/(platelets [109/L] × (ALT [U/L])^1/2^ were compared for the diagnosis of significant fibrosis and cirrhosis. FIB-4 proved to be more reliable than APRI in predicting significant fibrosis and cirrhosis. However, in a study published in 2016 [43], evaluating the performance of APRI and FIB-4 to predict fibrosis stage in 575 CHB, the majority (81–89%) of patients with advanced fibrosis or cirrhosis were missed by both scores. APRI and FIB-4 did not correlate with histologic fibrosis regression observed at five years of therapy. Recently, the performance of these two non-invasive scores for ruling out cirrhosis was evaluated using the established cutoffs to rule in (APRI > 2.00; FIB-4 > 3.25) or rule out (APRI < 1.00; FIB-4 < 1.45) cirrhosis and also aimed to identify new cutoffs for the task [44]. The data was acquired from eight global, randomized trials and two tertiary referral hospitals in the Netherlands and Canada. The study concludes that conventional cutoffs for APRI and FIB-4 should have high misclassification rates in HBV. A newly identified and externally validated cutoff for FIB-4 (≤0.70) was suggested to exclude cirrhosis in those patients over 30 years. The negative impact of age on the diagnostic performance and cutoffs of APRI and FIB-4 for significant fibrosis and cirrhosis in chronic hepatitis B was shown in a study of 2017 [45]. In 2011, Castera et al. published a study evaluating non-invasive tools, such as vibration-controlled transient elastography by Fibroscan^®^ (VCTE) (Echosens, Paris, France), Fibrotest^®^ (Biopredictive, Paris, France)—patented formula combining α-2-macroglobulin, γGT, apolipoprotein A1, haptoglobin, total bilirubin, age and gender—and APRI, for liver fibrosis assessment and follow-up in HBV HBeAg-negative chronic HBV infection [46]. VCTE (median 4.8 vs. 6.8 kPa, *p* < 0.0001), Fibrotest^®^ (0.16 vs. 0.35, *p* < 0.0001) and APRI values (0.28 vs. 0.43, *p* < 0.0001) were significantly lower in HBeAg-negative chronic HBV infection than in the remaining patients whereas they did not differ among HBeAg-negative chronic HBV infection according to HBV-DNA levels. In this study, over the time, significant fluctuations were observed for Fibrotest^®^ (median intra-patient changes at end of follow-up relative to baseline: +0.03, *p* = 0.012) and APRI (−0.01, *p* < 0.05) but not in VCTE (0.2 kPa, *p* = 0.12). In 2013, a study with 600 patients with CHB (36% HBeAg-negative chronic HBV infection) assessed fibrosis employing VCTE, FibroTest^®^, APRI, FIB-4 and liver biopsy (if indicated) [47]. They evaluated the incidence of death and liver transplantation during a 5-year follow-up and factors associated with overall survival. The overall survival was 94.1%, and survival without liver-related death was 96.3% (no liver-related death in HBeAg-negative chronic HBV infection). Survival was significantly decreased in patients diagnosed with severe fibrosis, whatever the non-invasive method used (*p* < 0.0001). In multivariate analysis, FibroTest^®^ and VCTE results had the highest hazard ratio with survival. Despite the good results found with serum markers and is a feasible option for assessing fibrosis, mainly a good option in locations with limited resources, liver elastography, mostly VCTE, has become the most validated method in CHB patients. The 2018 HBV Practice Guidance of the American Association for the Study of Liver Diseases recommends the use of elastography (preferred) or fibrosis biomarkers (FIB-4 or FibroTest^®^) to start therapy if these non-invasive tests indicate significant fibrosis [48].

### 3.2. Physical Tests

#### 3.2.1. Vibration-Controlled Transient Elastography by Fibroscan^®^

Due to the risk of hepatitis exacerbation or viral reactivation, more frequently observed in CHB than in HCV patients, the use of VCTE has been carefully analyzed. Irrespective of the hepatitis B e antigen (HBeAg) profile, factors that modify the elasticity of the liver, such as variations in the inflammatory infiltrate (flares or acute hepatitis), edema, vascular congestion, and cholestasis, could increase VCTE results. In these cases, the VCTE results must be evaluated with caution as measurements could not reflect liver damage but rather an inflammation [49,50,51,52]. Hence, elastography should not be performed in the presence of liver enzymes higher than five times the upper limit of normality (ULN) [49,53]. Some studies showed that, considering the ALT levels, the performance of VCTE is similar in viral hepatitis B and C [54,55]. The performance of liver elastography methods for detecting significant fibrosis in patients with chronic hepatitis B is superior to the serum markers [56,57]. VCTE is the most validated among the elastography methods, especially for diagnosing advanced fibrosis and cirrhosis [56]. Intermediate VCTE measurements have lower accuracy, and liver biopsy should be considered [53,58].

In HBeAg-negative chronic HBV infection, VCTE values showed fewer follow-up fluctuations than serum biomarkers as APRI and Fibrotest^®^, as demonstrated in the previously discussed Castera study [46]. In 2009, Maimone and cols. evaluated 220 CHB individuals (125 HBeAg-negative chronic HBV infection and 95 HBeAg-negative with persistently or intermittent elevation of ALT and/or HBV DNA > 105 copies/mL) [59]. The VCTE results, aminotransferase levels, and viral load are significantly lower in HBeAg-negative chronic HBV infection than those with HBeAg-negative chronic hepatitis B. VCTE showed to be an interesting tool to identify and follow those with inactive disease and to consider antiviral therapy [46,59].

In 2008, a study included 1197 HBeAg-negative treatment naïve CHB patients to investigate the relationship between HBV DNA and ALT levels and the risk of cirrhosis [60]. Possible and probable cirrhosis was defined as a VCTE result ≥ 8.4 kPa and ≥13.4 kPa and were described in 31% and 11% of the patients, respectively. The risk of cirrhosis was significantly increased when the ALT level was >0.5 × ULN or HBV DNA > 4 logs (10) copies/mL. Papatheodoridis et al. evaluated 357 untreated HBeAg-negative patients with VCTE: 182 HBeAg-negative chronic HBV infection with HBV-DNA < 2000 or 2000-19.999 IU/mL and 175 patients with CHB [61]. VCTE results did not differ between carriers with low and high viremia but were lower in carriers than in patients with CHB (5.8 ± 1.7 vs. 9.0 ± 5.6, *p* < 0.001), offering moderate differentiation between these two groups (AUROC 0.70). VCTE correlated significantly with grading and staging scores in patients with CHB and showed excellent accuracy for ≥moderate, ≥severe fibrosis or cirrhosis (AUROC ≥ 0.91–0.95).

In HBeAg-positive patients, older than 35 years and with normal ALT but close to the ULN, non-invasive assessment of fibrosis is convenient to differentiate the immune-tolerant phase from the presence of significant fibrosis secondary to clearance [62,63].

Several studies evaluated the accuracy of VCTE for fibrosis staging [53,56,57,58]. The variety of the population analyzed, and the prevalence of the disease can justify the differences in VCTE cutoff values identified. The European Association for the Study of the Liver (EASL)—sociación Latinoamericana para el Estudio del Hígado clinical practice guidelines for non-invasive tests, published in 2015, proposed an algorithm for the use of VCTE in treatment-naive patients with CHB [64]. Patients with normal ALT and VCTE values up to 5.9 kPa were considered without significant fibrosis, and with values > 9 kPa were classified as severe fibrosis/cirrhosis. The grey zone was defined as values between 6 and 9 kPa, and these patients should undergo, if possible, histological evaluation. Among patients with elevated transaminases (>1 to 5 times the ULN), those with VCTE values up to 5.9 kPa were considered without significant fibrosis and with values > 12 kPa were classified as severe fibrosis/cirrhosis. The grey zone was defined as values between 6 and 12 kPa, and these patients should undergo histological evaluation if possible.

The impact of therapy in CHB patients with advanced fibrosis and cirrhosis prior to treatment was evaluated in a study published in 2011 [65]. The VCTE used cutoffs was that proposed by Marcellin et al. (F2, F3 and F4 were ≥7.2 kPa, ≥8.1, and ≥11.0 kPa, respectively) [58]. Fifty-three patients (80%) had cirrhosis (F4), and 13 had (20%) advanced fibrosis (F3) prior to treatment. Among patients with F4 prior to treatment, 26 (49%) had liver stiffness below 11.0 kPa at follow-up. In F3 patients prior to treatment, 10 (77%) had liver stiffness below 8.1 kPa after treatment (median treatment duration was 50.5 months). In 2018, an Italian group evaluated 200 CHB patients and showed the positive impact of 24 months of therapy in VCTE measurements [66]. One hundred and forty-nine patients were treated with nucleos(t)ide analogs, while 51 patients were untreated. Patients with VCTE measurements ≥ 8.2 kPa were classified with advanced fibrosis (F ≥ 3). At baseline, the median VCTE value was 14.1 kPa for F ≥ 3 and 6.9 kPa for non-advanced fibrosis patients. The treated patients (68% received Entecavir and 32% Tenofovir) showed a decrease in VCTE measurement of 1.5 kPa (*p* < 0.001) in non-advanced fibrosis and of 6 kPa (*p* < 0.001) in F ≥ 3. In untreated patients, no statistically significant change of the VCTE was observed (*p* = 0.26). Even if VCTE results decreases with antiviral therapy, the decrease could not reflect the remission of liver fibrosis but inflammation improvement. A study that analyzed 556 CHB patients after 78-week entecavir-based therapy concluded that baseline Ishak fibrosis score was the only predictor of fibrosis reversion [67]. However, another prospective study from Asia also had a large cohort (*n* = 534) of CHB individuals, e antigen-positive treatment-naive patients receiving telbivudine-based therapy evaluated every six months with VCTE and compared with liver biopsies at baseline and week 104 [68]. The authors concluded that the early decline of 52-week VCTE measurements from baseline might reflect the remission of both liver inflammation and fibrosis and were predictive of 104-week fibrosis regression in treated patients with chronic hepatitis B.

#### 3.2.2. Ultrasound Elastography by Point Shear Wave and Two-Dimensional Shear Wave

Although VCTE is more validated for non-invasive assessment in CHB, ultrasound elastography by point shear wave (P-SW) and two-dimensional shear wave (2D-SW) have been increasingly evaluated through studies. As VCTE, Acoustic Radiation Force Impulse has better accuracy for cirrhosis diagnosis than significant fibrosis [69,70,71]. A combination of ARFI measurements with APRI and FIB-4 results to assess significant fibrosis might avoid some liver biopsies [72]. A meta-analysis in 2017 assessed the effect of ARFI elastography in the diagnosis of liver fibrosis in chronic hepatitis B and C and demonstrated high diagnostic accuracy for predicting advanced fibrosis and cirrhosis [73]. In this meta-analysis, ARFI cutoff values proposed for fibrosis staging in CHB were: 1.20 m/s for ≥F1, 1.45 m/s for ≥F2, 1.87 m/s for ≥F3, and 2.42 m/s for F4 [73]. A prospective study with 407 HBeAg-negative CHB patients evaluates the performance of P-SW to assess fibrosis [74]. VCTE, FIB-4, APRI, and FibroTest^®^ results were compared with P-SW on the same day and patients were followed for six years. P-SW results were significantly correlated with VCTE (r = 0.29, *p* < 0.001) and APRI (r = 0.17; *p* = 0.005). During six years of follow-up, median P-SW and VCTE values did not differ significantly over time (VCTE: *p* = 0.27; P-SW: *p* = 0.05).

A prospective multicenter study compared 2D-SW with other non-invasive methods (APRI, FIB-4, and VCTE) to evaluate 402 CHB patients. The authors observed higher accuracy of 2D-SW for ruling out and diagnosing cirrhosis [75]. A meta-analysis that evaluated the performance of 2D-SW in 13 sites (400 CHB patients) showed a good to excellent performance for the non-invasive staging of liver fibrosis in patients with hepatitis B and proposed the cutoff values for fibrosis staging in chronic hepatitis B: ≥7.1 kPa for ≥F2, ≥8.1 kPa for ≥F3, and ≥11.5 kPa for F4. Values < 8.4 kPa rule out cirrhosis [76].

In conclusion, staging liver fibrosis in chronic viral hepatitis is crucial, mainly to identify patients with advanced liver disease and implement strategies to improve prognosis and complications surveillance. Non-invasive tests are reliable tools to stratify fibrosis, and their use is extensively supported in daily clinics.

## Figures and Tables

**Table 1 viruses-14-00660-t001:** Characteristics of the Non-Invasive Tests used in the clinical practice for staging fibrosis in chronic hepatitis C.

Non-Invasive Test	Easy to Perform	Cost-Effective	Readly Available	High Diagnostic Accuracy for Advanced Fibrosis/Cirrhosis	Influenced by Biological Variation (Age, BMI)	Avoids Further Invasive or Other Complex Diagnostic Testing	False Positive Result
APRI	x	x	x	x (cirrhosis)	x **		
FIB-4	x	x	x	x	x **		
Fibrotest^®^		x		x	x **	x	x *
Fibrometer^®^					x **		
Hepascore^®^					x **		x *
ELF^®^		x		x	x **	x	
VCTE	x	x		x	x ^††^	x	
p-SWE				x	x ^†^	x	
2D-SWE				x	x ^†^		
MRE				x		x	

Legend: APRI, AST to platelet ratio index; FIB-4, Fibrosis-4; ELF, enhanced liver fibrosis score; VCTE, vibration-controlled transient elastography; p-SWE, point-shear wave elstography; 2D-SWE; Two-dimensional shear wave elastography; MRE, Magnetic Ressonance Elastography; * tests that uses bilirrubins may be false positive due to Gilbert Syndrome or haemolysis; ** age; ^†^ Body Mass Índex (BMI); ^††^ improved after XL probe development.

**Table 2 viruses-14-00660-t002:** Characteristics of the Non-Invasive Tests used in the clinical practice for staging fibrosis in chronic hepatitis B.

Non-Invasive Test	Easy to Perform	Cost-Effective	Readly Available	High Diagnostic Accuracy for Advanced Fibrosis/Cirrhosis	Influenced By Biological Variation (Age, BMI)	Avoids Further Invasive or Other Complex Diagnostic Testing	False Positive Result
APRI	x	x	x		x **		
FIB-4	x	x	x		x **		
Fibrotest^®^		x			x **	x	x *
VCTE	x	x		x	x ^††^	x	
p-SWE				x	x ^†^	x	
2D-SWE				x	x ^†^	x	

Legend: APRI, AST to platelet ratio index; FIB-4, Fibrosis-4; VCTE, vibration-controlled transient elastography; p-SWE, point-shear wave elstography; 2D-SWE; Two-dimensional shear wave elastography; * tests that uses bilirrubins may be false positive due to Gilbert Syndrome or haemolysis; ** age; ^†^ Body Mass Índex (BMI); ^††^ improved after XL probe development.

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
