# Peer review of "Staging Fibrosis in Chronic Viral Hepatitis"

_viruses, 2022, doi:10.3390/v14040660_

Round 1

Reviewer 1 Report

I read with great interest the review by Cardoso and al on fibrosis staging in chronic viral hepatitis. The manuscript is well written and very clear.

I have only minor suggestions:

  • please add a Conclusion section to your review.
  • page 2 line 72: please define CHC in the text
  • page 2 line 91: please define AUROC; line 92 "HIV-HCC co-infection", do you mean HIV-HCV co-infection ?
  • page 3 line 134: please define LSM
  • page 2 line 136-137: please indicate unit (kPa)
  • page 4 line 184: "low platelets > 150.000" - the word "low" is unsuitable
  • page 5 line 208, 212, 215 - all abbreviations have already been defined in the text

Author Response

We thank the Editor and the Reviewers of Viruses very much for their kind and valuable comments and suggestions that undoubtedly helped improve the quality of our manuscript. We have carefully addressed all Reviewers' recommendations. All changes in the manuscript are highlighted on a yellow background to facilitate the Reviewers. We hope that our manuscript is now suitable for publication in Viruses with these changes. If not, we are willing to go through further reviewing.

Reviewer reports:

Reviewer 1:

I read with great interest the review by Cardoso and al on fibrosis staging in chronic viral hepatitis. The manuscript is well written and very clear.

I have only minor suggestions:

Please add a Conclusion section to your review.

Thank you for this suggestion. We included a paragraph with the conclusion.

“In conclusion, staging liver fibrosis in chronic viral hepatitis is crucial, mainly to identify patients with advanced liver disease and implement strategies to improve prognosis and complications surveillance. Non-invasive tests are reliable tools to stratify fibrosis, and their use is extensively supported in daily clinics.

Page 2, line 72: please define CHC in the text

We apologize to the reviewer for this mistake. CHC was defined in the text as chronic hepatitis C, and the correction was highlighted (lines 42-43).  

Page 2, line 91: please define AUROC;

Thank you for this suggestion. We included the definition of AUROC (area under the receiving operating characteristic) in the text and maintained “AUROC” under parenthesis (lines 109-110).

Line 92, "HIV-HCC co-infection," do you mean HIV-HCV co-infection?

We apologize for this mistake again. Yes, the correct term is HIV-HCV. It was rewritten in the text and highlighted in yellow (line 112).

Page 3, line 134: please define LSM

Thank you for your observation. We defined LSM in the text as "liver stiffness measure" (line 160).

Page 2 line 136-137: please indicate unit (kPa)

Thank you. We included the unit after all cut-offs (line 164).

Page 4 line 184: "low platelets > 150.000" - the word "low" is unsuitable

Thank you, we excluded the word low (line 223-224).

Page 5 line 208, 212, 215 - all abbreviations have already been defined in the text

Thank you for this suggestion. We maintained only the abbreviations in the text since they have already been described. 

Reviewer 2 Report

The review by Cardoso et al. focuses on non-invasive tests for staging liver fibrosis in chronic hepatitis B and C. Eventually, the authors discussed the evaluation of liver fibrosis in HCV cured patients. The topic of the review is interesting and the presented data could be of use for a number of clinicians and researchers in the field of hepatology. However, in its present form, the review is somehow difficult to understand and I propose the following points to be addressed:

  1. The authors should have their manuscript reviewed for grammar, style, structure… The best is this to be done by a native English speaker. Some paragraphs/sentences are difficult to interpret.

An example is the sentence on line 131-135. Also, 22-24; 325-329, but there are more.

  1. A table of the abbreviations should be made available, since there are plenty of them. It should be clearly stated what CHC stands for. At some places the authors use CHC, on others, hepatitis C. Many other abbreviations are not introduced in the texts, like SVR (line 76), AST (line 84), ALT (line 86), AUROC (line 91) and so on.
  2. Headings/subheadings of the paragraphs concerning HCV and HBV non-invasive tests should be homogenous.

Currently, concerning HBV these are the headings:

Biological non-invasive tests

Indirect tests

Direct tests

Physical tests

And for HBV:

Noninvasive serum markers

Vibration-controlled transient elastography by Fibroscan

Ultrasound elastography by point shear wave and two-dimensional shear wave

This lack of consistency does not help the reader to find the proper classification of methods/markers.

  1. In order to help the readers, I would suggest the authors to make a table comparing the fibrosis markers/methods for HCV and HBV.
  2. Minor points that need attention:

Line 29: remove comma before brackets,

Line 40: “cytopathic effect of the hepatitis C virus”. This should be rephrased, since HCV does not cause typical CPE. CPE is usually referred to viruses causing cell lysis syncytia upon infection (like measles, HIV-1, etc).

Line 65: antiviral should be in plural

Line 78: I would remove “and without”, and also rephrase the sentence. The important point is to stratify those with decompensated cirrhosis.

Line 87-88:  I would change to “if a patient has a significant”

Line 89: change definition with issue for instance.

Line 92: “HIV-HCC coinfection” Maybe you meant HIV-HCV?

Line 97: change “have were” to have been

Line 102: Hyaluronic acid is a polysaccharide (GAG -glycosaminoglycan) and not a glycoprotein!

Line 169: “the values adopted before SVR should not be used”. Explain why.

Line 184: “low platelets >150 000/mm3”; it should be <150 000/mm3 (meaning less than)

Line 270: What do you mean by “inactive CHB”?

Line 290-293: This paragraph is unclear. Why would liver fibrosis be uncommon in HBeAg-positive patients?

Line 334 “has” should be “have”

Author Response

We thank the Editor and the Reviewers of Viruses very much for their kind and valuable comments and suggestions that undoubtedly helped improve the quality of our manuscript. We have carefully addressed all Reviewers' recommendations. All changes in the manuscript are highlighted on a yellow background to facilitate the Reviewers. We hope that our manuscript is now suitable for publication in Viruses with these changes. If not, we are willing to go through further reviewing.

Reviewer reports:

Reviewer 2:

Top of Form

The review by Cardoso et al. focuses on non-invasive tests for staging liver fibrosis in chronic hepatitis B and C. Eventually, the authors discussed the evaluation of liver fibrosis in HCV-cured patients. The topic of the review is interesting, and the presented data could be of use for a number of clinicians and researchers in the field of hepatology. However, in its present form, the review is somehow difficult to understand, and I propose the following points to be addressed:

The authors should have their manuscript reviewed for grammar, style, structure… The best is this to be done by a native English speaker. Some paragraphs/sentences are difficult to interpret.

An example is the sentence on line 131-135. Also, 22-24; 325-329, but there are more.

Thank you for the suggestion. The manuscript was reviewed carefully by a native English speaker, and the final version includes all the modifications.

A table of the abbreviations should be made available since there are plenty of them. It should be clearly stated what CHC stands for. At some places the authors use CHC, on others, hepatitis C. Many other abbreviations are not introduced in the texts, like SVR (line 76), AST (line 84), ALT (line 86), AUROC (line 91) and so on.

Thank you for your comment. In the text, we introduced the abbreviations in parenthesis the first time the word/term was mentioned [like chronic hepatitis C (CHC]. We did a review to keep the abbreviations all along with the text.

Headings/subheadings of the paragraphs concerning HCV and HBV non-invasive tests should be homogenous.

Currently, concerning HCV, these are the headings:

Biological non-invasive tests

Indirect tests

Direct tests

Physical tests

And for HBV:  

Non-invasive serum markers

Vibration-controlled transient elastography by Fibroscan

Ultrasound elastography by point shear wave and two-dimensional shear wave

This lack of consistency does not help the reader to find the proper classification of methods/markers.

Thank you for this suggestion. We made the changes and highlighted as suggested.

In order to help the readers, I would suggest the authors to make a table comparing the fibrosis markers/methods for HCV and HBV.

Thank you for this suggestion. We included the tables as suggested.

Minor points that need attention:

Line 29: remove comma before brackets,

We apologize for the mistake. We removed the comma.

Line 40: “cytopathic effect of the hepatitis C virus." This should be rephrased since HCV does not cause typical CPE. CPE is usually referred to viruses causing cell lysis syncytia upon infection (like measles, HIV-1, etc).

Thank you for your comment. We changed the whole paragraph and included a new reference.

Line 65: antiviral should be in plural

We corrected the word accordingly.

Line 78: I would remove “and without”, and also rephrase the sentence. The important point is to stratify those with decompensated cirrhosis.

We thank the reviewer for the suggestion. We removed the term “and without” and rephrased the sentence. The final sentence reads: “Furthermore, it is also necessary to stratify those with decompensating cirrhosis since this group will need additional stratification regarding treatment before or after liver transplant."

Line 87-88:  I would change to “if a patient has a significant”

Thank you for your comment. We made the change as suggested.

Line 89: change definition with issue for instance.

We thank the reviewer for this suggestion. We substituted “definition” with “issue.”

Line 92: “HIV-HCC co-infection” Maybe you meant HIV-HCV?

Thank you, we corrected the misspelling accordingly.

Line 97: change “have were” to have been

We apologized for this mistake and corrected it.

Line 102: Hyaluronic acid is a polysaccharide (GAG -glycosaminoglycan) and not a glycoprotein!

We thank the reviewer for this correction. We rewrote the phrase with the correct term.

Line 169: “the values adopted before SVR should not be used.” Explain why.

We thank the reviewers for the suggestion. We included the sentence as follows: “After SVR, there is a significant improvement in inflammation, and the results obtained from the NITs could reflect only this improvement and not of fibrosis”.

Line 184: “low platelets >150 000/mm3”; it should be <150 000/mm3 (meaning less than)

We apologize for this mistake and correct it promptly.

Line 270: What do you mean by “inactive CHB”?

Thank you for the question. We used the term (“inactive CHB”) proposed by the 2018 Practice Guidance of the AASLD for patients previously called inactive carriers. We changed this term for “HBeAg-negative chronic HBV infection”, as proposed in 2017 EASL HBV Guidelines.

Line 290-293: This paragraph is unclear. Why would liver fibrosis be uncommon in HBeAg-positive patients?

Thank you for your comment. We changed the paragraph as suggested.

Line 334 “has” should be “have”

Thank the reviewer for highlighting this mistake. We corrected it as asked (line 410).

Round 2

Reviewer 2 Report

The authors have completed the revision by integrating the suggested comments. In my opinion, the manuscript is now more comprehensive. The English is sufficiently good for understanding the report.